# PHLDA1 Does Not Contribute Directly to Heat Shock-Induced Apoptosis of Spermatocytes

**DOI:** 10.3390/ijms21010267

**Published:** 2019-12-30

**Authors:** Patryk Janus, Katarzyna Mrowiec, Natalia Vydra, Piotr Widłak, Agnieszka Toma-Jonik, Joanna Korfanty, Ryszard Smolarczyk, Wiesława Widłak

**Affiliations:** Maria Sklodowska-Curie Institute–Oncology Center, Gliwice Branch, Wybrzeże Armii Krajowej 15, 44-101 Gliwice, Poland

**Keywords:** apoptosis, detachment, heat shock, HSF1, PHLDA1, spermatogenesis

## Abstract

Spermatocytes are among the most heat-sensitive cells and the exposure of testes to heat shock results in their Heat Shock Factor 1 (HSF1)-mediated apoptosis. Several lines of evidence suggest that pleckstrin-homology-like domain family A, member 1 (PHLDA1) plays a role in promoting heat shock-induced cell death in spermatogenic cells, yet its precise physiological role is not well understood. Aiming to elucidate the hypothetical role of PHLDA1 in HSF1-mediated apoptosis of spermatogenic cells we characterized its expression in mouse testes during normal development and after heat shock. We stated that transcription of *Phlda1* is upregulated by heat shock in many adult mouse organs including the testes. Analyzes of the *Phlda1* expression during postnatal development indicate that it is expressed in pre-meiotic or somatic cells of the testis. It starts to be transcribed much earlier than spermatocytes are fully developed and its transcripts and protein products do not accumulate further in the later stages. Moreover, neither heat shock nor expression of constitutively active HSF1 results in the accumulation of PHLDA1 protein in meiotic and post-meiotic cells although both conditions induce massive apoptosis of spermatocytes. Furthermore, the overexpression of PHLDA1 in NIH3T3 cells leads to cell detachment, yet classical apoptosis is not observed. Therefore, our findings indicate that PHLDA1 cannot directly contribute to the heat-induced apoptosis of spermatocytes. Instead, PHLDA1 could hypothetically participate in death of spermatocytes indirectly via activation of changes in the somatic or pre-meiotic cells present in the testes.

## 1. Introduction

Elevated temperatures and other types of proteotoxic stresses lead to a Heat Shock Factor 1 (HSF1) activation, whose main targets are genes encoding molecular chaperones, primarily HSP proteins, providing cytoprotection [1]. However, not all cells respond to proteotoxic stress equally. The expression of inducible *Hsp70* genes is blocked in heat-shocked spermatocytes [2,3], while the expression of constitutively expressed testis-specific variants of HSP70 (HSPA2 and HSPA1L) is down-regulated after heat shock [4]. Furthermore, an over-expression of constitutively active HSF1 in mice leads to the apoptotic death of spermatocytes and male infertility [3,5,6]. Hence, spermatocytes and round spermatids are among the most heat-sensitive cells [7] and the most significant consequence of the heat stress in testes is the loss of germ cells via apoptosis [8]. Pleckstrin-homology-like domain family A, member 1 (PHLDA1) is activated in testes in the HSF1-dependent manner and heat-induced cell death has been diminished in the testes of PHLDA1-null mice [9]. Moreover, both HSF1 and PHLDA1 are expressed in cryptorchid rat testes in which apoptosis is induced leading to the loss of spermatogenic cells [10]. Therefore, it has been suggested that the upregulation of PHLDA1 by HSF1 could play a substantial role in promoting heat shock-induced cell death in spermatogenic cells.

PHLDA1 (pleckstrin-homology-like domain family A, member 1), also called TDAG51 (T-cell death-associated gene 51 protein), is an evolutionarily conserved proline-histidine and proline-glutamine rich protein broadly expressed in different tissues [11,12]. PHLDA1 expression can be modulated by a variety of stimuli, yet its precise physiological role is not well understood. It was identified as a potential stem cell marker [13] and has been suggested to play a role in tumorigenesis [14]. PHLDA1 expression was found to be upregulated in damaged skeletal muscle and its absence attenuated the early phases of muscle regeneration [15]. It plays a critical role in the development of progressive lung contusion and subsequent inflammation [16]. Furthermore, it is involved in the energy homeostasis by regulating lipogenesis in liver and white adipose tissue [17]. To date, several reports demonstrate that PHLDA1 may have either pro- [9] or anti-apoptotic [18,19] functions. It was induced upon T-cell activation–mediated apoptosis in vitro [20], yet PHLDA1-deficient mice displayed no apparent defects in T-cell apoptosis in vivo [21]. PHLDA1 was also shown to promote detachment-mediated cell death contributing to the development of atherosclerosis observed in hyperhomocysteinemia [22]. Nevertheless, apoptosis-related functions of PHLDA1 remain controversial. Aiming to elucidate potential role of PHLDA1 in the HSF1-mediated apoptosis of spermatogenic cells we characterized its expression in mouse testes after heat shock and during normal development.

## 2. Results

We tested the transcriptional induction of *Phlda1* in different mouse organs within 24 h of recovery after the heat shock and found that *Phlda1* transcripts were upregulated by hyperthermia in most organs (Appendix A). Importantly, the upregulation of *Phlda1* transcripts was also detected in mouse testes (Figure 1a). Moreover, the increased level of the PHLDA1 protein was detected by western blot in testes of mice subjected to heat shock (Figure 1b); it should be noted that the level of PHLDA1 was much lower in the testis (even after heat shock) than in the liver or NIH3T3 cells (two orders or one order of magnitude, respectively). To specify a type of spermatogenic cells that express *Phlda1*, we assessed its mRNA level during postnatal development starting from 11-day-old mice that already contain spermatogonia and the first leptotene spermatocytes in seminiferous tubules [23] (a scheme in Figure 1c). We compared the expression of *Phlda1* and genes with different spermatogenesis-related patterns: *Hspa2*, which expression starts from leptotene spermatocytes with the highest level in pachytene spermatocytes [24], *Pgk2*, with the most abundant expression in pachytene spermatocytes from around the 15th day of development [25], and *Dazl*, which is expressed in spermatogonia but not in early spermatocytes and later stages [26]. We found that the *Phlda1* transcript level only slightly rose from spermatogonia and leptotene spermatocytes (11–13-day-old mice) through early and late pachytene spermatocytes (15- and 18-day-old mice, respectively) to the stage corresponding to round spermatids appearance (21-day-old males); this pattern generally resembled one characteristic for *Dazl* (Figure 1d,f). A similar developmental pattern of the PHLDA1 protein was observed when whole tissue lysates were analyzed by western blot using two different antibodies (for specificity tests, see Appendix A). PHLDA1 was already detected in 11-day-old mice and its level started to decrease in 25-day-old mice; it is noteworthy that the appearance of pachytene spermatocytes (15- and 18-day-old mice) did not result in the increased level of PHLDA1 (Figure 1e). In parallel, we analyzed mice that expressed a mutated constitutively active form of HSF1 specifically in spermatocytes which led to their apoptosis. As a result, testes of such mice contained only spermatogenic cells at earlier developmental stages (i.e., spermatogonia and early spermatocytes) but no spermatids and spermatozoa [3,6]. In testes of transgenic mice, the level of *Phlda1* mRNA was similar to its level in wild-type animals (except 13-day-old animals when a slightly higher level of *Phlda1* transcript was detected in transgenic males) (Figure 1f). A similar observation was made for PHLDA1 protein: the higher level of PHLDA1 was detected only in testes of 13-day-old transgenic males then similar levels of this protein were observed in testes of older wild-type and transgenic animals (Figure 1g). In marked contrast, transcript levels of *Hspa2* and *Pgk2* genes (specific for pachytene spermatocytes) in these animals were significantly lower than in wild-type animals, apparently because of apoptotic death of pachytene spermatocytes [3]. On the other hand, the expression of the spermatogonia-specific *Dazl* gene was similar in wild-type and transgenic animals (Figure 1f). Presented data indicate collectively that *Phlda1* becomes expressed in mouse testes at early developmental stages which correspond to the appearance of spermatogonia and/or leptotene spermatocytes, yet its products do not accumulate significantly in the later stages of testicular development.

Furthermore, we performed immunofluorescence analyses using anti-PHLDA1 antibodies to show the localization of this protein in the specific compartment of the mouse testes. This analysis indicated that PHLDA1 barely accumulated in spermatogenic cells and its expression was observed mostly in somatic cells present in testes (between seminiferous tubules, putatively Leydig cells and pericytes), while pachytene spermatocytes revealed only very weak staining (Figure 2a). Moreover, PHLDA1 did not accumulate after heat shock in any type of spermatogenic cells through the apoptosis was induced and DNA breaks were detected 4–6 h after the treatment, especially in spermatocytes (Figure 2b). On the other hand, in agreement with a previously published report [10], PHLDA1 accumulated in cryptorchid testes 14 days after surgery, also in spermatogenic cells. However, the accumulation of apoptotic cells in cryptorchid testes could be detected earlier than PHLDA1 up-regulation (7 days, but not 14 days after surgery; Figure 2c; see also [7]). This indicates the lack of correlation between heat-induced apoptosis and PHLDA1 accumulation in mouse spermatocytes.

To further study mechanisms of hypothetical PHLDA1-mediated heat-induced cell death, mouse NIH3T3 cells were transiently transfected with vectors coding for PHLDA1 and PMAIP1; the latter protein is a known apoptosis-inducing factor [27]. Overexpression of the PMAIP1/EGFP (enhanced green fluorescent protein) or EGFP/PMAIP1 fusion protein led to the activation of caspase-3 and -7. In marked contrast, overexpression of the PHLDA1/EGFP or EGFP/PHLDA1 fusion protein only slightly activated caspase-7, but not caspase-3, compared to control cells expressing EGFP (Figure 3a). Moreover, phenotypes of NIH3T3 cells transiently transfected with the above vectors were analyzed by live-cell imaging microscopy. PMAIP1/EGFP overexpression induced apoptosis and green cells disappeared from the field of view 2–3 h after characteristic membrane blebbing lasting for 40–60 min (Figure 3b). On the other hand, the overexpression of EGFP/PHLDA1 (very effectively produced after only a few hours of transfection) and PHLDA1/EGFP (not so effectively produced possibly due to lack of the Kozak sequence in the original PHLDA1 translation start site) led to the detachment of cells, yet classical apoptosis was not observed (Figure 3c). These results indicate that PHLDA1 and PMAIP1 induce alternative mechanisms of cell elimination.

## 3. Discussion

We revealed that *Phlda1* transcription begins in the early stages of mouse testicular development and its protein products do not accumulate in later developmental stages of spermatogenic cells. Hayashida and coworkers [9] detected *Phlda1* mRNA in spermatocytes by in situ hybridization, though they were unable to show protein expression in these cells. Nevertheless, these authors suggested that PHLDA1 could play a substantial role in promoting heat shock-induced cell death in mouse testes and based on in vitro studies proposed a pro-apoptotic mechanism of PHLDA1 action. However, analysis of PHLDA1-induced cell death was not apoptosis-specific in the above study because cells adhering and detaching from dishes were simply counted leaving room for other types of cell death (e.g., anoikis). HSF1/PHLDA1 pathway was also suggested to have a role in cell death of primary spermatocytes in cryptorchid rat testes [10]. However, in this model, the highest level of apoptosis was observed before the accumulation of the PHLDA1 protein. Importantly, we found that PHLDA1 protein does not accumulate in spermatocytes and round spermatids, which are the most heat-sensitive cells in testes [7] where apoptosis is induced in HSF1-dependent manner [3,5]. Therefore, it is unlikely that PHLDA1 could contribute directly to any apoptosis-related processes in heat-sensitive spermatocytes as was suggested in some previous studies.

Nevertheless, we noted that heat shock slightly increased *Phlda1* expression in mouse testes. It was previously reported that increased PHLDA1 expression could promote cell detachment [22,28], which was confirmed in our study. We also found that PHLDA1 overexpression did not activate caspase-3, which is the effector caspase necessary for apoptosis. It is noteworthy, however, that overexpression of PHLDA1 could activate caspase-7 which has no significant role in sensitivity to intrinsic cell death, but it is responsible for cell detachment [29]. Therefore, it is possible that PHLDA1 may contribute to the death of testicular cells at elevated temperatures. However, its potential role in heat-induced apoptosis of spermatocytes could be indirect and could involve damage and death of somatic cells present in testes that support functions of spermatogenic cells. Germ cell adhesion to Sertoli cells is crucial for maintaining the integrity of the seminiferous tubule and it was shown that loss of Sertoli-germ cell adhesion determines the rapid germ cell elimination during the seasonal regression of the seminiferous epithelium occurring in seasonal breeders [30]. One can assume that PHLDA1 leading to cell detachment may be involved in germ cell loss by similar mechanisms. Hence, the role of hypothetical PHLDA1-mediated mechanisms involved indirectly in the death of spermatogenic cells deserves further studies, also in the context of stress-related male infertility.

## 4. Materials and Methods

### 4.1. Animals and Heat Shock Treatment

Adult (10–16-week-old), inbred FVB/N male mice were used for heat shock treatments. For the whole-body heat treatment, mice were anesthetized with i.p. injection of avertin (15–17 µL of a 2.5% solution/g body weight), and then the lower half of the torso of each animal was submerged in a water bath at 43 °C for 30 min. For experimental cryptorchidism, we applied the procedure described elsewhere [31]. We also employed FVB/N juvenile wild-type and transgenic males expressing a mutated, constitutively active transcriptionally-competent form of HSF1 specifically in spermatogenic cells [3,6] and Balb/c males (11–46-day-old). For each experimental point, three males were used. The animal experiments were carried out according to Polish legislation and were approved by the Local Committee of Ethics and Animal Experimentation at the Medical University of Silesia in Katowice, Poland (Decisions No 82/2009 made on 25 November 2009 and No 129/2014 made on 17 December 2014). 

### 4.2. RNA Isolation, cDNA Synthesis, and (q)PCR

Total RNA was isolated using the GeneMATRIX Universal RNA Purification Kit (Eurx, Gdańsk, Poland) or The Ambion^®^ PARIS™ kit (#AM1921, ThermoFisher Scientific, Waltham, MA, USA), digested with DNase I (Worthington Biochemical Corporation, Lakewood, NJ, USA), converted into cDNA and used for RT-PCR as described [32]. Quantitative PCR (qPCR) was performed as described elsewhere [33]. Each reaction was performed in triplicate in two independent experiments. The set of delta-Cq replicates for control and test samples were normalized according to the geometric mean of the reference *Gapdh* housekeeping gene. Normalized values were used for expression fold change calculation using the double delta Cq method (by determining the median, maximum and minimum value) and for estimation of the *p*-values. The primers used in these assays are described in Appendix A.

### 4.3. Transient Transfections, Protein Extraction, and Western Blotting

Mouse NIH3T3 cells plated one day before the experiment were transfected with plasmid DNA using X-tremeGENE™ HP DNA Transfection Reagent (Sigma-Aldrich, St. Louis, MO, USA) or polyethylenimine (PEI, Sigma-Aldrich; 3:1 ratio of PEI:DNA) according to the instructions provided by the producer. Cells were harvested 24 h later if not specified otherwise. The cellular proteins were extracted using either Radio-Immunoprecipitation Assay (RIPA) buffer consisting of 1× PBS, 1% Nonidet-40, 0.5% sodium dodecyl cholate, 0.1% sodium dodecyl sulfate, 1 mM phenylmethylsulfonyl fluoride (PMSF), 50 mM NaF, protease inhibitors cocktail, and phosphatase inhibitors PhosStopTM (Roche, Indianapolis, IN, USA) or buffer consisting of 187 mM Tris-HCl, pH 6.8, 2% SDS, 0.05% Triton X-100, 10% glycerol, 1 mM PMSF, 1mM EGTA, and protease inhibitors cocktail (Roche, Indianapolis, IN, USA). Testis samples were homogenized on ice and disrupted by sonication (15 MHz, three pulses of 30 s each). The lysed samples were centrifuged at 14,000× *g* for 10 min at 4 °C and the supernatant was saved as protein extract. The Ambion^®^ PARIS™ kit (#AM1921, ThermoFisher Scientific, Waltham, MA, USA) was used for analyses of transgenic mice testes. Proteins (30–50 µg) were separated on SDS-PAGE gels and electrotransferred to 0.45-μm pore nitrocellulose filters (Merck Millipore, Burlington, MA, USA). There were following primary antibodies used: anti-PHLDA1 (1:200–1:500; *Ab1*, mouse monoclonal, sc-23866, Santa Cruz Biotechnology, Santa Cruz, CA, USA, or *Ab2*, rabbit polyclonal, NBP1-84969, Novus Biologicals, Littleton, CO, USA), anti-GFP (1:200; sc-8334, Santa Cruz Biotechnology, Santa Cruz, CA, USA), anti-caspase-3 (1:3000; #14220), anti-cleaved caspase-3 (1:1000; #9664), anti-caspase-7 (1:3000; #12827), and anti-cleaved caspase-7 (1:1000; #8438), from Cell Signalling Technology (Danvers, MA, USA). As a loading control, anti-ACTB (1:10,000; MAB1501, Merck Millipore, Burlington, MA, USA) or anti-HSPA8 (1:5000; sc-7298, Santa Cruz Biotechnology Santa Cruz, CA, USA) were run using the same blot if only possible. The primary antibody was detected by an appropriate secondary antibody conjugated with horseradish peroxidase (1:1000–1:5000) and visualized by WesternBright ECL kit (Advansta, Menlo Park, CA, USA). Imaging was performed on x-ray film or in G:BOX chemiluminescence imaging system (Syngene, Frederick, MD, USA). Blots were subjected to densitometric analyses using Image Studio Lite software version 5.2.5 (LI-COR Biosciences, Lincoln, NE, USA) to calculate relative protein abundance with a loading control normalization (levels of statistical significance were calculated using the T-test).

### 4.4. Immunofluorescence (IF) and TUNEL Assay

Reactions were performed on sections (6 µm) of formalin-fixed (10% in PBS, overnight at 4 °C) and paraffin-embedded mouse testes. An antigen retrieval step in 0.01 M citrate buffer pH 6.0 was performed before procedures. IF imaging was performed using a primary antibody against PHLDA1 (1:100; NBP1-84969, Novus Biologicals, Littleton, CO, USA (*Ab2*) and Alexa Fluor 647 Tyramide SuperBoost Kit goat anti-rabbit IgG (#B40926, Invitrogen, Waltham, MA, USA) according to the user guide. This kit offers sensitivity 10–200 times higher than that of standard immunostaining methods. Staining with mouse anti-PHLDA1 (1:50–1:100; sc-23866, Santa Cruz Biotechnology, Santa Cruz, CA, USA (*Ab1*) was done using Mouse on Mouse Fluorescein Kit (#FMK-2201, Vector Laboratories, Burlingame, CA, USA). Apoptotic cells with DNA breaks were detected using In Situ Cell Death Detection Kit, TMR red (#12 156 792 910, Roche, Indianapolis, IN, USA) according to the supplier’s protocol. Finally, the DNA was stained with DAPI. Images were captured using Zeiss Axio Imager.M2 fluorescent microscope with AxioCam camera and AxioVision (Rel.4.8) imaging or using Zeiss confocal microscope LSM710, AxioObserver (Carl Zeiss Microscopy GmbH, Jena, Germany) and ZEN software. Negative controls were performed in parallel on serial slides for specific labeling by omitting the primary antibody.

### 4.5. Phlda1 and Pmaip1 Cloning

Mouse *Phlda1* coding sequence was amplified by PCR on DNA template and cloned into *PstI* site in pEGFP-C1 vector (Takara/Clontech, Saint-Germain-en-Laye, France) to obtain EGFP/PHLDA1 or into *PstI* site in pEGFP-GL3 vector (created by replacing *Luc* in pGL3 control vector, Promega, Madison, WI, USA, with *EGFP*) to obtain PHLDA1/EGFP (stop codon was replaced by Leu codon) using In-Fusion^®^ HD Cloning Kit (Takara/Clontech, Saint-Germain-en-Laye, France) according to the user manual. The mouse *Pmaip1* promoter and 5′UTR (up to –686 from ATG) along with coding sequence were amplified by PCR on DNA and cDNA from testes, respectively, as templates. *Pmaip1* stop codon was removed (TGA to TGG) and the *NcoI* site was created in reverse primer enabling in-frame ligation with *Egfp* coding sequence from the pEGFP-1 vector (Takara/Clontech, Saint-Germain-en-Laye, France). *Pmaip1* and *Egfp* sequences were inserted to a backbone of the pGL3-Enhancer vector (Promega, Madison, WI, USA) between *KpnI* and *XbaI* sites to obtain construct encoding the PMAIP1/EGFP fusion protein under the control of the *Pmaip1* promoter. To obtain EGFP/PMAIP1, the mouse *Pmaip1* coding sequence was amplified by PCR and cloned into the *PstI* site in the pEGFP-C1 vector using In-Fusion^®^ HD Cloning Kit (Takara/Clontech, Saint-Germain-en-Laye, France) according to the user manual. The DNA sequence of the recombinant plasmids was confirmed by sequencing (Genomed, Warszawa, Poland).

### 4.6. Life Cell Imaging

Cells were plated onto Nunc^®^ Lab-Tek^®^ II chambered coverglass (#155383, Nalge Nunc International, Rochester, NY, USA) or on 96-well plate one day before the experiment. Plasmid DNA was transfected to mouse NIH3T3 cells using X-tremeGENE™ HP DNA Transfection Reagent (Sigma-Aldrich, St. Louis, MO, USA) according to supplier’s recommendations. Next day cells were transferred to the humidified chamber (37 °C, 5% CO_2_) on the microscope stage. Axio SD Cell Observer Z1 (Carl Zeiss Microscopy GmbH, Jena, Germany) microscope was used with EC-Plan Neofluar 40×/1.30 Oil DIC M27 objective or 10× objective, 488 nm excitation and 509 nm emission wavelength for the green fluorophore. Image capture was performed using the Zeiss software.

## Figures and Tables

**Figure 1 ijms-21-00267-f001:**
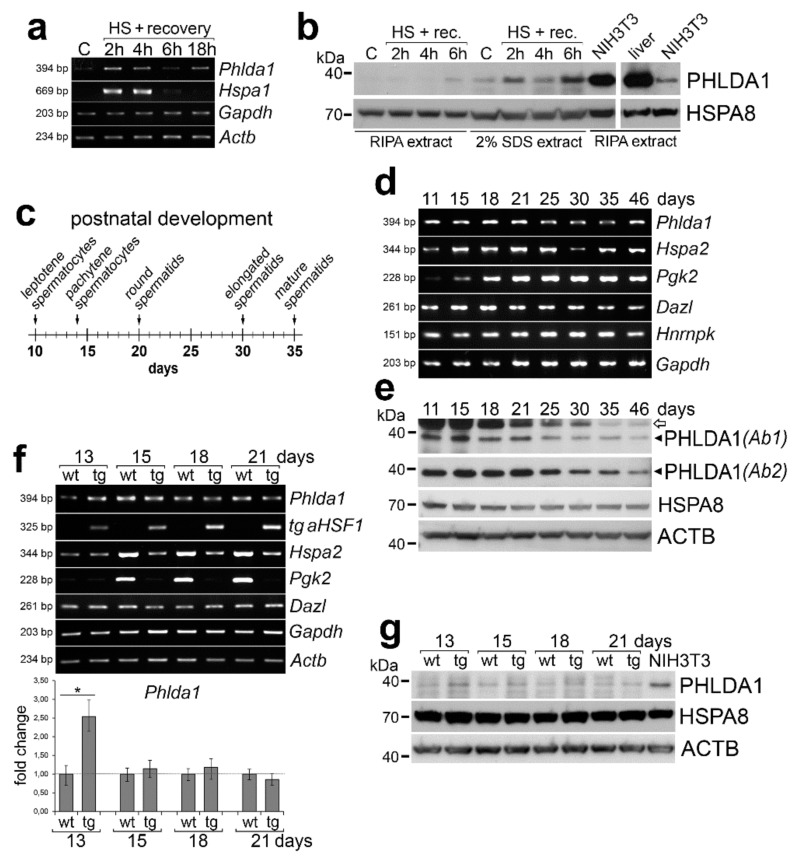
Expression of pleckstrin-homology-like domain family A, member 1 (PHLDA1) in mouse testes. (**a**) Transcripts of *Phlda1* and reference genes analyzed by RT-PCR in adult testes after heat shock performed in vivo and indicated recovery time. C—control, physiological temperature; HS—heat shock. (**b**) PHLDA1 protein level analyzed by western blot in testes of mice subjected to heat shock and indicated recovery time. HSPA8 was used as loading control; proteins were extracted with either RIPA or 2% SDS buffer. (**c**) Time-line of the appearance of different spermatogenic cells during the mouse postnatal development. (**d**) Transcripts of *Phlda1* and reference genes analyzed by RT-PCR in testes of 11–46-day-old animals. (**e**) PHLDA1 protein level analyzed by western blot in testes of 11–46-day-old animals. ACTB was used as loading control; two anti-PHLDA1 antibodies (*Ab1* and *Ab2*) were used; heavy chain IgG detected by the secondary anti-mouse antibody is marked with an arrow. (**f**) Transcripts of *Phlda1* and reference genes analyzed by RT-PCR in testes of wild-type (wt) and aHSF1 transgenic (tg) mice at 13th, 15th, 18th, and 21st day of postnatal development (upper panel); fold change in *Phlda1* expression quantified by RT-qPCR in testes of tg mice compared to wt mice of the same age (bottom panel; marked are minimum and maximum values). Asterisk indicates the statistical significance of differences (* *p* < 0.05). (**g**) PHLDA1 protein level analyzed by western blot in testes of wild-type (wt) and aHSF1 transgenic (tg) mice. ACTB and HSPA8 were used as loading controls.

**Figure 2 ijms-21-00267-f002:**
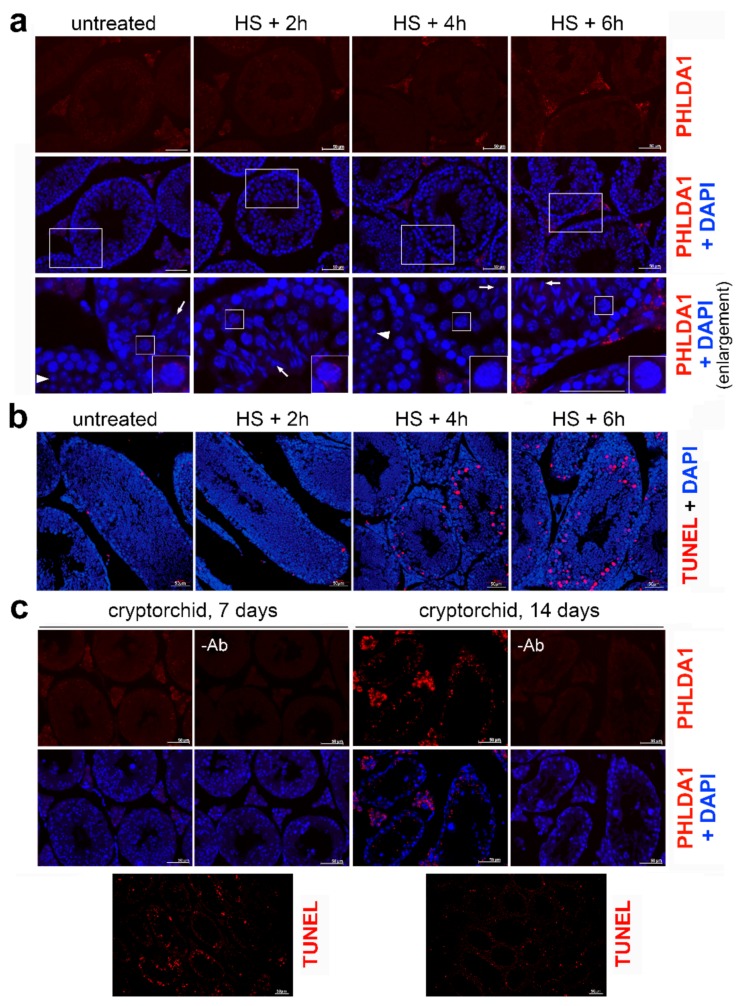
Localization of PHLDA1 in mouse testes. (**a**) Detection of PHLDA1 by immunofluorescence (using *Ab2*, red; DNA stained with DAPI, blue) in testes of a control mouse and subjected to heat shock after 2, 4, and 6 h of recovery. Enlargement of the marked areas is shown in the bottom panel; round spermatids and condensing spermatids are marked with arrowhead and arrows, respectively. Representative pachytene spermatocytes (in squares) are further enlarged in the lower corners. (**b**) Detection of apoptotic DNA breaks (by TUNEL assay, red; DNA stained with DAPI, blue) in seminiferous tubules of untreated mice and after heat shock in vivo and indicated recovery time (2–6 h). (**c**) Detection of PHLDA1 by immunofluorescence (red; DNA stained with DAPI, blue) in cryptorchid testes 7 and 14 days after surgery (upper panel) and detection of apoptotic DNA breaks (by TUNEL assay, red) in corresponding tissues (bottom panel). Scale bar—50 µm.

**Figure 3 ijms-21-00267-f003:**
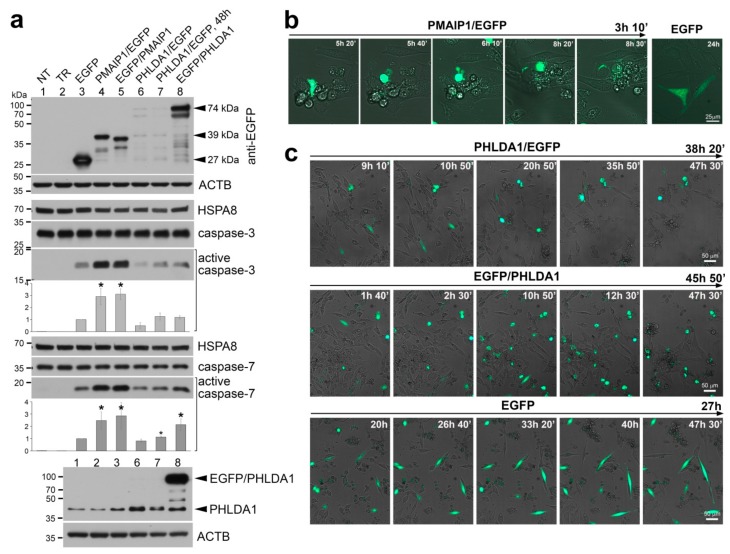
Mouse PHLDA1 can induce caspase-7 (but not caspase-3) activation and detachment of cells. (**a**) Mouse NIH3T3 cells were transiently transfected with vectors coding for indicated fusion proteins and protein extracts were analyzed by western blot. NT—not transfected cells, TR—transfection reagent only. ACTB and HSPA8 were used as loading controls. Active caspase-3 and caspase-7 were quantitated based on densitometry of replicated blots and results are shown in corresponding charts; asterisks indicate statistically significant (* *p* < 0.05) differences against the EGFP-transfected cells. (**b**) An example of the live-cell imaging of NIH3T3 cells transiently transfected with the PMAIP1/EGFP construct showing the apoptotic membrane blebbing in a cell overexpressing PMAIP1, while control cells (transfected with the EGFP vector only, right photo) were not affected. (**c**) An example of the live-cell imaging of NIH3T3 cells transiently transfected with vectors coding for PHLDA1/EGFP, EGFP/PHLDA1 or EGFP showing detachment of cells overexpressing PHLDA1. Images were recorded in EGFP (green) and brightfield channels for two days. The observation period shown for each vector is given above the panels. The relative time of acquisition is displayed in the upper right corner of each image.

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
