# Peer review of "PHLDA1 Does Not Contribute Directly to Heat Shock-Induced Apoptosis of Spermatocytes"

_ijms, 2019, doi:10.3390/ijms21010267_

Round 1

Reviewer 1 Report

Smart research about indirect influence of PHLDA1 on heat shock-induced apoptosis of spermatocytes.

There is no aim in abstract and in the end of introduction.

It is necessary to provide in the section of Discussion some information about future practical implementation of fundamental research results 

Author Response

Referee #1:

Comment 1. There is no aim in abstract and in the end of introduction.

Answer 1. The aim of the study was identified in the revised abstract and introduction, accordingly.

Comment 2. It is necessary to provide in the section of the Discussion some information about future practical implementation of fundamental research results.

Answer 2. The last paragraph of the revised Discussion was modified to indicate possible directions for future studies.

Reviewer 2 Report

The paper is potentially interesting, and the results relevant to research in the field of causes of male infertility. However, before accepting the manuscript, a better description of the quantitative RT PCR should be included. The authors state that they used qPCR, but all the results presented only show gel electrophoresis, and no table of actual fold increase is available. This should be rectified. The authors should include a table detailing the qRT PCR numerical results.

Secondly, the IF in figure 2 (localisation of PHLDA1 in mouse tested) show images at very low magnification. While this allows to visualise the cross section of a whole tubule, it would be important to include higher magnification images, to  detect if any pachytene cell contain low level of protein.

Finally, the difference in blebbing between NIH3T3 overexpressing PHLDA1 or PMAIP1 is not sufficient evidence to support the claim that the two proteins induce alternative methods of cell elimination. Other markers of apoptosis vs necrosis vs autophagy should be used, in IF or WB experiments.

Author Response

Referee #2:

Comment 1. The paper is potentially interesting, and the results relevant to research in the field of causes of male infertility. However, before accepting the manuscript, a better description of the quantitative RT PCR should be included. The authors state that they used qPCR, but all the results presented only show gel electrophoresis, and no table of actual fold increase is available. This should be rectified. The authors should include a table detailing the qRT PCR numerical results.

Answer 1. We apologize that qPCR results were not described properly. During the revision, we have modified the relevant figure (Figure 1f) and figure description to meet the standards of such analyses. Although most RT-PCR analyses were performed semi-quantitatively and results are shown as gel electrophoresis, we decided to re-analyze the Pmaip1 expression by qPCR in wild-type and transgenic mice testes to quantify the potential Pmaip1 gene up-regulation by transgenic, active HSF1. Each qPCR reaction was performed in triplicate in two independent experiments and average results are shown in Fig. 1f (bottom panel) as fold-changes of transgenic versus age-matched control animals.      

Comment 2. Secondly, the IF in figure 2 (localisation of PHLDA1 in mouse tested) show images at very low magnification. While this allows to visualise the cross section of a whole tubule, it would be important to include higher magnification images, to  detect if any pachytene cell contain low level of protein.

Answer 2. Figure 2 has been changed accordingly during revision and tubules cross-sections have been enlarged (bottom PHLDA1+DAPI panel). Representative pachytene spermatocytes are also shown as additional enlarged inserts. Nonetheless, PHLDA1 is hardly detected in the untreated mouse testes even using Tyramide SuperBoost Kit (which offers sensitivity 10–200 times higher than that of standard ICC/IHC/ISH methods). However, using extended image acquisition time, weak staining could be noticed in pachytene spermatocytes in normal testes which disappeared 4-6 hours after heat shock treatment. The clear PHLDA1 staining is visible only in cryptorchid testes.

Comment 3. Finally, the difference in blebbing between NIH3T3 overexpressing PHLDA1 or PMAIP1 is not sufficient evidence to support the claim that the two proteins induce alternative methods of cell elimination. Other markers of apoptosis vs necrosis vs autophagy should be used, in IF or WB experiments.

Answer 3. The main goal of our work was to verify the hypothetical role of PHLDA1 in HSF1-mediated apoptosis of spermatogenic cells. We found that PHLDA1 is hardly detected in the most heat-sensitive spermatogenic cells in untreated mouse testes and not detectable in pachytene spermatocytes undergoing HSF1-mediated, heat-induced apoptosis (i. e. 4-6 hours post-treatment). This indicated that PHLDA1 lacks an essential direct role in heat-induced apoptosis of spermatocytes. Moreover, the overexpression of PHLDA1 in NIH3T3 cells did not result in the activation of caspase-3, an established apoptotic marker. In marked contrast, the overexpression of PMAIP1, which has a well-established role in the induction of apoptosis, resulted in the activation of caspase-3 and membrane blebbing, which are characteristic features of apoptosis. Different types of cell death are usually characterized by distinguished morphological features yet knowledge of molecular markers specific for a given death phenotype is frequently limited. Hence, it is important to note that cells overexpressing PHLDA1 seemed to be eliminated without morphological (membrane blebbing) and molecular (activation of caspase-3) features of apoptosis. We agree that more studies are necessary to characterize mechanisms induced by PHLDA1. Hence, our conclusion on a hypothetical role of PHLDA1 in putative death mechanisms mediated by cell detachment is only speculative and requires further studies. Nevertheless, we documented that overexpression of PHLDA1 resulted in the activation of caspase-7, which was previously postulated to be involved in cell detachment (Brentnall et al. 2013). Moreover, it was reported that PHLDA1 upregulation resulted in cell shape change and β-catenin nuclear translocation in human renal proximal tubules (Carlisle et al. 2012). Germ cell adhesion to Sertoli cells is crucial to maintain the integrity of the seminiferous tubule and it was shown that loss of Sertoli-germ cell adhesion determines the rapid germ cell elimination during the seasonal regression of the seminiferous epithelium occurring in seasonal breeders (Luaces et al. 2014). One can assume that PHLDA1 leading to cell detachment may be involved in germ cell loss by similar mechanisms. However, a systemic analysis of putative PHLDA1-mediated mechanisms hypothetically involved in the rearrangement of the seminiferous epithelium was beyond the scope of current communication and relevant work could not be completed upon a short time (five days) given for revision.

References

Brentnall M, Rodriguez-Menocal L, De Guevara RL, et al (2013) Caspase-9, caspase-3 and caspase-7 have distinct roles during intrinsic apoptosis. BMC Cell Biol 14:32. https://doi.org/10.1186/1471-2121-14-32

Carlisle RE, Heffernan A, Brimble E, et al (2012) TDAG51 mediates epithelial-to-mesenchymal transition in human proximal tubular epithelium. Am J Physiol Renal Physiol 303:F467-481. https://doi.org/10.1152/ajprenal.00481.2011

Luaces JP, Rossi LF, Sciurano RB, et al (2014) Loss of Sertoli-germ cell adhesion determines the rapid germ cell elimination during the seasonal regression of the seminiferous epithelium of the large hairy armadillo Chaetophractus villosus. Biol Reprod 90:48. https://doi.org/10.1095/biolreprod.113.113118